# High-speed secure random number generator co-processors for privacy-preserving machine learning

**Shannon Egan**[1]

[1]Deep Science Ventures , shannon@deepscienceventures.com

## Abstract

As machine learning (ML) increasingly handles sensitive data, there is growing need for secure implementations of privacy-preserving techniques like differential privacy (DP). While random number generation is essential for ML applications, from basic operations to advanced privacy mechanisms, current solutions face a critical trade-off: modern pseudo-random number generators (PRNGs) are highly optimized for ML workloads, but lack the cryptographic guarantees required for secure real-world DP implementations. Our benchmark of private training shows a 43-530% increase in single-step runtime after switching to a cryptographically-secure generator–even with available hardware acceleration. This result highlights a major gap in integration of secure RNGs into GPU-accelerated ML. In this position paper, we argue that dedicated hardware RNG co-processors could bridge this gap by providing high-throughput true random numbers from physical entropy sources while dramatically reducing power consumption compared to software implementations. Such co-processors would be especially valuable for on-device private learning and other edge AI applications where both security and energy efficiency are essential.

## 1 Introduction

Innovation in hardware acceleration and software optimization for machine learning (ML) have enabled significant advancement in the scale and efficiency of ML models. With these advancements, ML applications can be practically deployed in new use cases, from training on large sensitive datasets (1; 2) to on-device ML training and inference (3; 4).

Differential privacy (DP) (5; 6) has emerged as a leading framework for privacy-preserving computations on sensitive data, with organizations like Google, Apple, and the US Census Bureau adopting DP for both standard database calculations and differentially-private ML (DP-ML) (7; 8; 9). Injecting random noise into statistical calculations is the fundamental feature of DP protocols; the added noise limits the amount of information that can be extracted about individual entries or subsets of the sensitive dataset. But implementing truly secure DP-ML requires more than just adding noise to training operations—the noise itself must also be unpredictable. This creates a fundamental tension between security and performance: while modern ML frameworks achieve excellent performance using fast pseudo-random number generators (PRNGs)(10; 11), these algorithms lack the cryptographic guarantees needed for secure DP implementations. Now that DP models are deployed to production by the likes of Apple and Google for processing private data on mobile devices, their security is of great concern for consumer privacy (12; 13; 14).

In this position paper, we explore how a high-speed secure RNG co-processor could interface with existing ML frameworks and accelerators to enable privacy-preserving applications with-

out sacrificing performance. Such a co-processor would support ML workloads by generating cryptographically-secure random numbers from a physical entropy source, and feeding them into applications running on CPUs/GPUs through high-speed interconnects. We believe a high-performance true random number generator (TRNG) implementation is achievable in current CMOS technology, with significantly lower power consumption than software alternatives. This technology could have near-term impact on secure ML deployment, especially on power-constrained edge and mobile devices.

In Section 2 we review the current state of random number generation in ML frameworks and highlight a significant gap in support for cryptographically-secure RNGs in GPU-accelerated applications. Section 3 presents benchmark results for performance of differentially-private training with cryptographically-secure noise generation across different architectures and models. Our results show dramatic training step time increases when using a cryptographically-secure RNG, even with hardware acceleration enabled. Finally, Section 4 outlines design requirements and integration approaches for a high-speed TRNG co-processor that could address these performance challenges.

## 2   Secure RNG for differential privacy

Differentially-private ML (DP-ML) training (15), or private training, has become the standard for building ML models from sensitive data. DP provides a mathematical framework for quantifying and limiting how much information about any individual training example can be extracted from the trained model, tracked through a cumulative "privacy budget" that measures total information leakage. Most private training implementations are based on the DP-stochastic gradient descent (DP-SGD) optimization method (16), in which carefully calibrated random noise is added to gradient updates during training.

DP-ML practitioners and cryptography experts consistently emphasize the need for cryptographically-secure PRNGs (CSPRNGs) to generate the random noise (17; 18; 19; 20; 21), as insecure PRNGs may leave the system vulnerable to attacks which invalidate the privacy budget.

Non-CSPRNGs are vulnerable to state compromise extension attacks, whereby an attacker that has discovered the PRNG's internal state can clone the generator and reproduce the past and/or future sequence of pseudorandom numbers (22). For many insecure PRNG designs, observing a relatively small number of pseudorandom outputs can allow an attacker to computationally reconstruct or guess the internal state of the PRNG. Many practical implementations of this attack on the Mersenne Twister (23), the default PRNG in Python and PyTorch CPU (24; 25), have been documented (18; 26; 27).

In the case of DP-ML, where noise is added to sparse gradient vectors, an attacker could infer generator outputs even if they do not have direct access to them. The zero elements of the gradient vectors (typically returned as intermediate variables during training execution and therefore can be leaked) expose information about the pseudorandom sequence which could be used to discover the generator state. Garfinkel et al. (18) describe an analogous attack on sparse DP Census data. A successful state compromise extension attack on the PRNG used for noise addition could then enable data reconstruction attacks on the DP-model (28; 29; 30). Such attacks render the information-theoretic limits specified by the privacy budget obsolete.

Floating point attacks have also been extensively studied in DP-ML applications (31; 32; 33), but effective mitigation strategies are now widely adopted (33; 34; 35; 36).

Despite these weaknesses, PRNGs continue to be used in privacy applications because the performance tradeoff for existing hardware sources of randomness are too extreme (18). In most practical settings, a CSPRNG seeded by a true random source, provided on most modern processors and accessible via /dev/urandom on UNIX-based systems, is considered an acceptable compromise of performance and security (17; 18; 19).

Given the strong consensus on the necessity of CSPRNGs for secure DP implementations, we were surprised to find that none of the 3 most popular DP-ML libraries—TensorFlow Privacy (37),

PyTorch Opacus[1] (38) , JAX Privacy (39)—support noise generation with a CSPRNG. Nor does NVIDIA's cuRAND library for random number generation implement a single CSPRNG (10).

There is thus a clear gap for integration of cryptographic-strength RNGs (either pseudo or true) into production-ready libraries for GPU-accelerated applications, including but certainly not limited to DP-ML. This gap persists despite the fact that CUDA implementations of stream ciphers like Advanced Encryption Standard (AES) for cryptographically-secure random number generation have long been reported in the literature (40; 41; 42; 43; 44). These have demonstrated throughput up to 867 Gbps (44)—over an order of magnitude faster than the $\sim$10 Gbps throughput typically given by CSPRNGs on high-performance CPUs (45), but still far behind the blistering speeds up to 1 TBps[2] achieved with insecure counter-based PRNGs like Philox (41) on NVIDIA A100 GPUs (10).

To quantify the performance impact of using CSPRNGs in private training, we implemented DP-SGD with both standard PRNGs and CSPRNGs. Our benchmark reveals significant runtime overhead that could be mitigated through hardware acceleration.

## 3 DP-SGD Benchmark

### 3.1 Implementation

Many references in the DP literature and codebases of private training frameworks mention a steep performance cost when training DP applications with noise generated by CSPRNGs (19; 38), but the impact is rarely quantified. To better understand this problem, we designed a DP-SGD benchmark which measures the contribution of secure noise generation to runtime of a single private training step.

We evaluated the impact of cryptographically-secure random number generation on two representative neural network architectures: 1) ResNet-18 (46) as implemented in torchvision (47), and 2) a small Transformer encoder model consisting of 2 layers with 4 attention heads, input dimension 128, and hidden dimension 256; both implemented as binary classifiers. These architectures were chosen to represent both traditional computer vision workloads and modern attention-based models which have become ubiquitous in ML.

Private training was implemented using Opacus' PrivacyEngine, which provides highly optimized implementations of DP-SGD on PyTorch. The PrivacyEngine was configured with 1.0 noise multiplier and 1.0 L2 gradient clipping norm, and uses the Poisson subsampling scheme (48) with average batch size of 32.

Due to the lack of built-in CSPRNGs in existing ML libraries, we were forced to implement a custom PyTorch generator for cryptographically-secure random number generation. We follow a common implementation (41; 25; 44) of the stream cipher AES to generate cryptographically-secure random bitstreams. Our Generator leverages the Python cryptography library (49) for hardware-accelerated AES, utilizing Intel AES-NI instructions on x86 processors (50) and the Secure Enclave on Apple Silicon (51). To minimize overhead, the generator draws and caches batches of 10M random numbers at a time. When the cache is exhausted, a new batch is generated using fresh entropy from the system RNG. The hardware-accelerated AES Generator showed strong performance in raw random number generation tests, achieving similar runtime or even modest speedup over the default PyTorch generator on CPU (up to $1.2\times$ for tensor size $> 1$M elements). We note that this AES generator is a demonstration for the purpose of assessing performance of CSPRNGs in private training, and has not been rigorously tested for its statistical quality and security.

The benchmark compares training step times between two configurations of the PrivacyEngine: one using our AES-based Generator for noise generation, and one with noise explicitly disabled by setting the noise multiplier to zero. This allows us to isolate the overhead specifically due to secure noise generation while accounting for other DP-SGD operations like gradient clipping and per-sample gradient computation. We also tested the benchmark with the default Opacus PrivacyEngine

---

[1]Technically Opacus supports CSPRNG via torchcsprng (25; 21), but we encountered major compatibility issues between the outdated csprng package (last updated 2021) and modern PyTorch/CUDA versions. Because of this, we implemented our own secure Generator for the benchmark in Section 3.

[2]Note the difference in unit convention for performance metrics (bits vs. bytes per second). We opt for consistency with cited sources rather than converting.

Table 1: Model runtime comparison with training operation breakdown for DP-SGD training step with AES CSPRNG noise generation. We use a custom Transformer Encoder model described in Section 3.1, and torchvision's ResNet-18 (47) with 1.6M and 11M trainable parameters, respectively. All runtimes reported in ms with mean and standard deviation calculated over 1000 DP-SGD step iterations. Note that noise generation is a subset of the DP-SGD update time.

| Transformer Encoder | | | | | | |
|---|---|---|---|---|---|---|
| System | Apple Silicon M3 | | CoLab VM T4 | | CoLab VM A100 | |
| Noise setting | On | Off | On | Off | On | Off |
| Forward/backward pass | $43. \pm 10.$ | $43. \pm 10.$ | $17. \pm 3.$ | $17. \pm 3.$ | $13.6 \pm 0.6$ | $14. \pm 1.$ |
| DP-SGD update | $57. \pm 38.$ | $19. \pm 8.$ | $45. \pm 37.$ | $6. \pm 3.$ | $44. \pm 33.$ | $6. \pm 2.$ |
| (Noise generation) | $(39. \pm 5.)$ | $(2.4 \pm 5)$ | $(39. \pm 6.)$ | $(0.7 \pm 0.2)$ | $(38. \pm 2.)$ | $(0.7 \pm 0.1)$ |
| Total step time | $100. \pm 40.$ | $63. \pm 15$ | $61. \pm 37.$ | $22. \pm 4.$ | $58. \pm 34.$ | $20. \pm 3.$ |
| Noise overhead | 58% | | 174% | | 186% | |

| ResNet-18 | | | | | | |
|---|---|---|---|---|---|---|
| System | Apple Silicon M3 | | CoLab VM T4 | | CoLab VM A100 | |
| Noise setting | On | Off | On | Off | On | Off |
| Forward/backward pass | $478. \pm 175.$ | $473. \pm 173.$ | $37. \pm 6.$ | $38. \pm 6.$ | $31. \pm 4.$ | $31. \pm 2.$ |
| DP-SGD update | $326. \pm 219.$ | $89. \pm 29.$ | $268. \pm 235.$ | $15. \pm 5.$ | $234. \pm 195.$ | $10. \pm 4.$ |
| (Noise generation) | $(239. \pm 11.)$ | $(5. \pm 2.)$ | $(253. \pm 42.)$ | $(1.3 \pm 0.4)$ | $(223. \pm 6.)$ | $(1.18 \pm 0.07)$ |
| Total step time | $804. \pm 291.$ | $562. \pm 184.$ | $306. \pm 235.$ | $53. \pm 9.$ | $265. \pm 195.$ | $42. \pm 4.$ |
| Noise overhead | 42% | | 475% | | 528% | |

and PyTorch Generator, but as expected the PyTorch PRNG was not a bottleneck for private training, with noise overhead reaching at most 2%.

We evaluated the benchmark on three different systems to understand how noise generation overhead changes with available compute resources: An Apple Silicon M3 system with 8 CPU cores and Metal Performance Shaders (MPS) backend for PyTorch (52); and Google Colab VMs with 2 Intel Xeon CPU cores and 1 NVIDIA GPU (either T4 or A100).

All runtime measurements were averaged over 1000 DP-SGD training step iterations. The code for our DP-SGD benchmark is available at the git repo `https://github.com/sm-egan/rng_ml`, including example Jupyter notebooks suitable for CoLab.

## 3.2  Results

The benchmark results in Table 1 reveal substantial overhead from our cryptographically-secure noise generation across all systems tested. Enabling secure noise generation for the Transformer model increased total step time by 58% on the M3, 174% on the T4, and 186% on the A100. The noise generation component dominates DP-SGD update time, accounting for 68-89% of the update

across all systems. Notably, while GPU acceleration reduces runtime of the forward/backward pass and other DP-SGD overheads significantly, the noise generation remains CPU-bound, limiting the overall benefit of more powerful accelerators.

The overhead becomes even more pronounced with the larger ResNet-18 model. Absolute noise generation time scales by $\sim 6\times$ relative to the Transformer Encoder, roughly proportionate to the model size increase (11M trainable parameters vs. 1.6M). The CUDA GPUs accelerate runtime on non-noise operations by more than $10\times$ compared to the M3, yet the runtime contribution of noise generation is relatively constant, yielding an astounding $\sim 500\%$ noise overhead. The fact that noise generation times are similar across systems, even with a significant difference in CPU core count, suggests either that our AES CSPRNG implementation is not taking advantage of parallel-processing or multi-threading on CPU; or that GPU-CPU memory transfer is the predominant bottle-neck regardless of raw noise generation performance. Disentangling these effects and parallelizing the AES CSPRNG are important directions for future work in order to make relevant performance comparisons to other parallel RNGs.

These results again highlight the tradeoffs between security and performance that practitioners face when implementing secure RNGs for DP-ML, and suggest two key directions for future work to facilitate adoption of privacy-preserving ML:

1. Efficient integration of optimized software CSPRNGs and their hardware entropy sources into common software frameworks for DP-ML, and

2. Increasing throughput while maintaining randomness quality in hardware entropy sources and TRNGs, allowing them to play a bigger role in generating truly unpredictable noise for security-critical applications.

In the following section, we discuss the limitations of current TRNG implementations and propose guidelines for new technologies to bridge the security-performance gap.

## 4 The path to RNG co-processors for privacy-preserving ML

Some DP practitioners, including the US Census Bureau, have expressed their preference for a reliable hardware implementation of secure RNG (18). Specialized hardware TRNGs can generate random numbers with superior power efficiency and security, as the internal RNG state is not initialized in memory, but in an actual physical process that the TRNG circuit implementation merely samples (53). The entropy source may be thermal or electronic noise, jitter in ring oscillator circuits (54; 55; 56), or a quantum process such as single photon emission and detection (57). TRNGs are complex systems which require careful design to ensure their quality and security.

TRNGs are already commercialized for application in cryptography, but operate at only moderate speeds (up to 200-500 Mbps) (58; 59) and produce random bit or integer outputs. Most commercial TRNG solutions consist of IP Cores that can be implemented on FPGAs, but the difficulties of FPGA programming and integration often hamper their adoption.

These existing TRNGs are unlikely to meet the performance and integration demands of DP-ML applications. New designs for commercially-viable RNG co-processors are needed, integrating a significantly higher-speed entropy source with additional logic units to convert bitstream outputs to continuous distributions useful for DP-ML: uniform, Gaussian, Laplacian, and Poisson in particular. To facilitate integration with this logic, the TRNG should be implemented in CMOS-compatible hardware with existing high-yield manufacturing processes. This would enable chip designs that serve as efficient co-processors for secure noise generation in DP workloads, analogous to Apple's Secure Enclave cryptography co-processor for keygen and encryption tasks. Unlike general purpose hardware, such co-processors can be designed with specific memory and logic architectures that mitigate the DP noise attacks described in Section 2.

We believe the most promising hardware platform for high-speed TRNG cores are emerging non-volatile memory (NVM) technologies (60). These devices are low power, often CMOS-compatible, and randomness can be harvested from the stochastic behaviour observed when switching the memory state. Because of these properties, NVMs have been studied extensively for application in neuromorphic computing and stochastic computing (61; 62). With respect to designing a high-speed TRNG, the key advantage of NVMs is that individual memory cells can be assembled into large

arrays with multiple cells accessible in parallel. The array size and degree of parallelism can be optimized during design in order to produce a device with higher random bit throughput, even if access time is fixed.

To facilitate integration with other parts of the ML workflow one could deliver the RNG co-processor as an add-on card with PCI Express (PCIe) interface, the same standard interface used by GPUs, or integrate it as a chiplet in a Sytem on Chip. The latter approach is in keeping with industry trends towards chiplet architectures for more flexible adaptation to changing AI workloads (63).

Based on performance benchmarks for PRNG and our survey of existing TRNG technologies, we propose the following design guidelines for an RNG co-processor:

1. Random bit generation at $>100\,\mathrm{Gbps}$

2. Energy efficiency on the order 1-10 picojoules-per-bit ($\mathrm{pJ/bit}$)

3. Modular component or chiplet connecting to system through standard interfaces (PCIe, UCIe, etc.)

4. Compliant with NIST guidelines for random number generation SP 800-90A/B/C (64; 65; 66), and able to pass standard statistical tests for binary sequences such as NIST 800-22 (67) and TestU01 (68).

This aspirational target of 100 Gbps is set with the goal of approaching data rates achieved in high-performance PRNGs, and matching data transfer rates of standard interfaces like PCIe Gen 6.0, which sits at 121 GBps in the maximum 16-lane configuration. The planned PCIe Gen 7.0 is expected to double the maximum data transfer rate to 242 GBps by 2025.

Energy efficiency of $1\text{-}10\,\mathrm{pJ/bit}$ is routinely achieved in TRNGs (69; 70), but would be unimaginable for even for the most efficient CUDA implementations of CSPRNGs. Efficiency values reported in the literature range from $0.43\text{-}4\,\mathrm{Gbps/W}$ (42; 44), equivalent to 245-2326 $\mathrm{pJ/bit}$.

## 5 Discussion

The benchmark presented in Section 3 is an early attempt to quantify the oft-cited security vs. performance tradeoff in current DP-ML infrastructure. Hardware accelerators and software optimization have dramatically improved the performance of private training computations, but cryptographically-secure RNGs remain a bottleneck for applications with stricter security requirements, thus limiting the adoption of truly secure DP.

Our findings show that even hardware-accelerated AES on CPU is not sufficient to achieve reasonable performance for secure noise generation. CPU-GPU transfers are a well-known bane to ML performance, and we believe these effects contribute to the 43-530% overheads for CSPRNG noise generation observed in our benchmark. Implementing GPU-accelerated CSPRNGs in common DP-ML libraries is an important next step for the field to ensure that applications built on these frameworks can maintain their privacy guarantees. This solution should suffice in the data centre setting, but the high power cost of CSPRNG implementations on GPUs may be impractical in resource-constrained environments like edge devices and mobile applications, where DP-ML is often deployed.

Looking ahead, we see the development of secure RNG co-processors as a critical enabler for privacy-preserving ML. As models continue to scale and privacy regulations become more stringent, the ability to efficiently generate high-quality random numbers for DP-ML and related technologies such as federated learning and secure aggregation (71; 72; 35; 73) will become even more important. New hardware solutions could help bridge the gap between the theoretical guarantees of DP and practical, deployable implementations that preserve both privacy and performance.

## Acknowledgments

The author extends thanks to Dr. Paolo Toccacelli for helpful discussions, to Dr. Brock Doiron for invaluable feedback on earlier drafts, and to the Advanced Research and Invention Agency (ARIA) for supporting broader foundational work in this space.

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
