# OpenReview forum: "High-speed secure random number generator co-processors for privacy-preserving machine learning"
_NeurIPS.cc/2024/Workshop/MLNCP — MLNCP Poster_

### Official Review · Reviewer_FxEK · 2024-09-26
**This position paper proposes using TRNG co-processors to improve ML/AI accelerator performance; however, there is relatively little discussion of the current impact of PRNGs on ML/AI performance.**

**Rating:** 6
**Confidence:** 3

**Review:**

This position paper proposes using TRNG co-processors to improve ML/AI accelerator performance.  It is a compelling idea that could improve future AI/ML systems, particularly those that will require significant random sampling.  This position paper could help influence the design of future AI/ML accelerator decisions.

Pros:

-- Compelling idea, with good initial motivation

-- Quantifiable recommendations provided, helping to guide future work

Cons:

-- I would like to have seen more quantification of the current impact of PRNGs on AI/ML performance to help motivate this position.

-- I would like to have seen more description on how and why the guidelines were chosen.

---

### Decision · Program_Chairs · 2024-10-10

Accept (Poster)